# GEODESIC MODE CONNECTIVITY

**Charlie Tan,**[*] **Theodore Long,**[*] **Sarah Zhao**[*] **& Rudolf Laine**[*]
University of Cambridge
`{ct632,tl559,smxz2,lrl34}@cam.ac.uk`

## ABSTRACT

Mode connectivity is a phenomenon where trained models are connected by a path of low loss. We reframe this in the context of Information Geometry, where neural networks are studied as spaces of parameterized distributions with *curved geometry*. We hypothesize that shortest paths in these spaces, known as *geodesics*, correspond to mode-connecting paths in the loss landscape. We propose an algorithm to approximate geodesics and demonstrate that they achieve mode connectivity.

## 1 INTRODUCTION

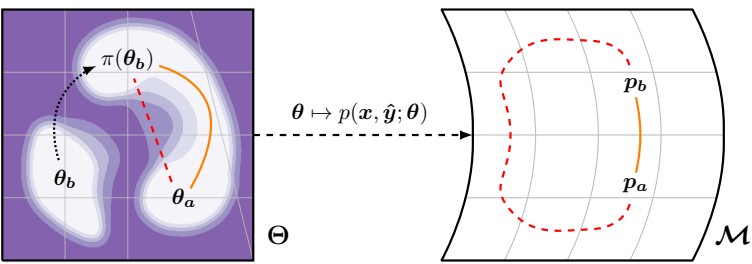

Figure 1: **Geodesics are shortest paths in the space of parameterized distributions $\mathcal{M}$.** For narrow architectures linear interpolation (dashed) fails to achieve mode connectivity, passing through a region of high loss, despite using a permutation $\pi$ to 'shift' $\boldsymbol{\theta_b}$ closer to $\boldsymbol{\theta_a}$ (Ainsworth et al., 2022). If we instead follow the geodesic (shortest) path (solid) in the curved distribution space, this *does* achieve mode connectivity, appearing as a curved path in the loss landscape.

Garipov et al. (2018) demonstrated the existence of **mode connectivity**, wherein stochastic gradient descent (SGD) solutions are connected by low-loss paths in the loss landscape. This result challenged the prevailing perspective of isolated minima in favour of large connected valleys. More recently, Ainsworth et al. (2022) achieved **linear mode connectivity** (LMC), finding *linear* paths between two minima with no increase in loss. They achieved this by exploiting the natural permutation symmetries of neural network layers (Entezari et al., 2022) to 'shift' one of the models into the same loss basin as the other, enabling LMC (see Figure 1, LHS).

Information Geometry interprets neural networks as parameterized distributions $p(\boldsymbol{x}, \hat{\boldsymbol{y}}; \boldsymbol{\theta})$, studying them using tools from differential geometry (Amari, 2012). In this setting, the distribution space $\mathcal{M}$ is endowed with a *metric* which determines lengths and gives the space a curvature, much like Einstein's theory of general relativity defines a metric describing curved spacetime. One can then define **geodesics** as the *paths of shortest length* between two points in this space.

Our work demonstrates a connection between mode connectivity and geodesics in the curved distribution space. In particular, **we hypothesize that all geodesics between SGD solutions are mode-connecting paths**. Our specific contributions include:

1. A novel algorithm for approximating geodesic paths through the loss landscape.
2. Using our algorithm to find mode-connecting paths between narrow ResNets trained on CIFAR-10, where existing methods require wider architectures.

---

[*]equal contributions

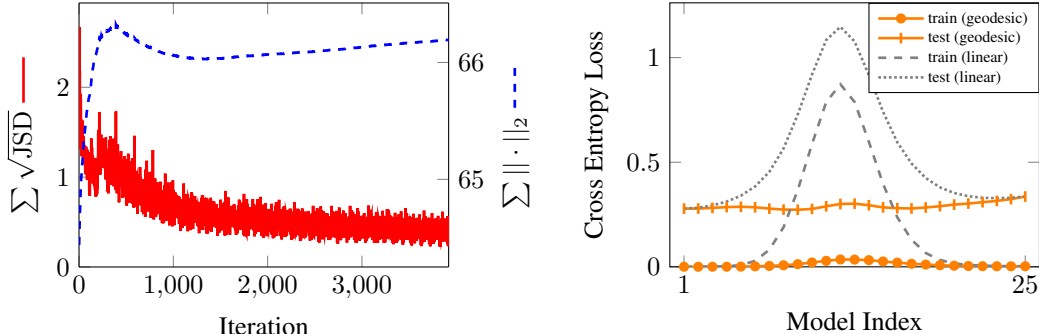

Figure 2: **Geodesic optimization achieves mode connectivity for ResNet20 on CIFAR-10. Left: Geodesic optimization minimizes discretized length functional**. Our algorithm finds a curved path in parameter space; although this curvature increases the Euclidean length (dashed) compared to the linear path, it decreases the distribution space length (solid). **Right: Geodesic optimization identifies low-loss paths where linear mode connectivity has failed**. Linear interpolation between $\boldsymbol{\theta_1}$ and $\boldsymbol{\theta_N}$ shows a large increase in loss on both train and test datasets. After geodesic optimization, all points on the path have low loss, which we refer to as **geodesic mode connectivity**.

## 2 GEODESIC OPTIMIZATION

Given two trained neural networks, $\boldsymbol{\theta_a}$ and $\boldsymbol{\theta_b}$, we seek to approximate the geodesic path $\gamma$ between the distributions $\boldsymbol{p_a} = p(\boldsymbol{x}, \hat{\boldsymbol{y}}; \boldsymbol{\theta_a})$ and $\boldsymbol{p_b} = p(\boldsymbol{x}, \hat{\boldsymbol{y}}; \boldsymbol{\theta_b})$. This path minimizes the length functional defined in Equation 1, where $g_{ij}$ is the Fisher-Rao metric (see Appendix A.1). This functional is equivalent to the integral of the infinitesimal square root Jensen-Shannon Divergence (JSD) (Crooks, 2007). Our algorithm uses a discrete version of Equation 1. We first initialize a sequence of $N$ models $\{\boldsymbol{\theta_a} = \boldsymbol{\theta_1}, \boldsymbol{\theta_2}, \ldots, \boldsymbol{\theta_N} = \boldsymbol{\theta_b}\}$ along a linear path. Keeping the endpoints fixed, we optimize parameters $\{\boldsymbol{\theta_i}\}$ to minimize the loss defined in Equation 2. Note two differences with Equation 1: (i) We approximate the integral as a discrete sum, measuring the JSD between distributions defined by each $\boldsymbol{\theta_i}$, similar to (Carter et al., 2008) (ii) We do not take the square root of JSD; this is equivalent to minimizing the *energy* functional which gives the *unique* geodesic of constant velocity.

$$L(\gamma) = \int_t \sqrt{\frac{d\gamma^i}{dt} g_{ij} \frac{d\gamma^j}{dt}} dt = \sqrt{8} \int_\gamma \sqrt{d\text{JSD}} \tag{1}$$

$$\mathcal{L}(\{\boldsymbol{\theta_i}\}_{i=1}^N) = \sum_{i=1}^{N-1} \text{JSD}(\boldsymbol{p_i} || \boldsymbol{p_{i+1}}) \tag{2}$$

## 3 EXPERIMENTS

We conduct experiments with a $4 \times$ width ResNet20 architecture (He et al., 2016) on the CIFAR-10 dataset (Krizhevsky, 2009). We independently train two networks, $\boldsymbol{\theta_a}$ and $\boldsymbol{\theta_b}$, and use the weight matching algorithm of Ainsworth et al. (2022) to permute $\boldsymbol{\theta_b}$ closer to $\boldsymbol{\theta_a}$. Even after permuting, the linear path between the two models still shows a large increase in loss, as seen in Figure 2 (RHS). We then use the algorithm defined in Section 2 with $N = 25$. Note this only requires training images, no labels or test data are employed. See Appendix A.2 for further details. The results in Figure 2 demonstrate that our algorithm achieves mode connectivity, where LMC has failed.

## 4 CONCLUSION

Geodesic optimization succeeds in identifying paths of low-loss between narrow ResNets, where linear mode connectivity is not present. Future work could compare geodesic paths with those found by the method of Garipov et al. (2018), or theoretically investigate these geodesic paths.

ACKNOWLEDGEMENTS

The authors thank Dr Ferenc Huszár and Dr Challenger Mishra for their support in defining this project, and feedback on our submission. The authors further thank Mr Oisin Kim and Mr Yonatan Gideoni for their feedback on our writing.

URM STATEMENT

All authors are first-time submitters, and meet the URM criteria of ICLR 2023 Tiny Papers Track.

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

# A  APPENDIX

## A.1  NEURAL NETWORK INFORMATION GEOMETRY

In the context of supervised learning for classification tasks, a neural network with parameters $\boldsymbol{\theta}$ can be interpreted as a conditional distribution $p(\hat{\boldsymbol{y}}|\boldsymbol{x};\boldsymbol{\theta})$, which maps an input $\boldsymbol{x}$ to a probability distribution over a set of class labels $\hat{\boldsymbol{y}}$. Coupled with a distribution over the inputs $p(\boldsymbol{x})$, we get a parameterized joint probability distribution $p(\boldsymbol{x},\hat{\boldsymbol{y}};\boldsymbol{\theta}) = p(\boldsymbol{x})p(\hat{\boldsymbol{y}}|\boldsymbol{x};\boldsymbol{\theta})$. In practice, we take $p(\boldsymbol{x})$ to be the empirical distribution of our input data to obtain the joint probabilities discussed in the main text.

This space of joint distributions, under certain assumptions, is a Riemannian manifold equipped with a Riemannian metric known as the **Fisher-Rao metric $g$**. This is also known, in broader contexts, as the Fisher information matrix. Given a parameter vector $\boldsymbol{\theta} = (\theta_1, \ldots, \theta_d)$, where $d$ is the total number of parameters, the $(i, j)$-coordinate of the Fisher-Rao metric is given by Equation 3. For further details on information geometry and its application to neural networks, see Amari (2012).

$$g_{ij}(\boldsymbol{\theta}) = \mathbb{E}_{p(\boldsymbol{x}, \hat{\boldsymbol{y}}; \boldsymbol{\theta})} \left[ \frac{\partial \log p(\boldsymbol{x}, \hat{\boldsymbol{y}}; \boldsymbol{\theta})}{\partial \theta_i} \frac{\partial \log p(\boldsymbol{x}, \hat{\boldsymbol{y}}; \boldsymbol{\theta})}{\partial \theta_j} \right], \tag{3}$$

## A.2 IMPLEMENTATION DETAILS

### A.2.1 RESNET ARCHITECTURE

The ResNet architecture employed by Ainsworth et al. (2022) and this work differs from the standard ResNet He et al. (2016) by the use of LayerNorm (Ba et al., 2016) instead of BatchNorm Ioffe & Szegedy (2015). This is due to BatchNorm layers lacking invariance to permutations. The use of LayerNorm was observed to decrease the test accuracy of the models by a few percentage points compared to BatchNorm.

### A.2.2 EXPERIMENTAL PROCEDURE

We train two separate models $\boldsymbol{\theta_a}$ and $\boldsymbol{\theta_b}$ from different random seeds using SGD. We then employ the weight matching algorithm of Ainsworth et al. (2022) to permute $\boldsymbol{\theta_b}$, accounting for the permutation symmetries of the parameter space. We next linearly interpolate between $\boldsymbol{\theta_a}$ and $\pi(\boldsymbol{\theta_b})$ to obtain $N$ models $\{\boldsymbol{\theta_i}\}_{i=1}^N$, where $N = 25$. Keeping the end points fixed, we optimize the parameters $\boldsymbol{\theta_i}$ using SGD with a learning rate of 0.1 and a batch size of 256, to minimise the loss as given by Equation 2.

