# OpenReview forum: "Geodesic Mode Connectivity"
_ICLR.cc/2023/TinyPapers — Submitted to Tiny Papers @ ICLR 2023_

### Official Review · Reviewer_tP1F · 2023-03-27

**Confidence:** 3

**Summary Of Contributions:**

The paper proposes to investigate connections between different weights values for the same neural network architecture. It proposes an algorithm to approximate a geodesic interpolation between them, avoiding high-loss regions. The method is tested on ResNet20, showing similar cross-entropy loss.

**Rating:**

Clear, Correct, and Reproducible (CCR): a submission which meets the reviewing criteria

**Strengths And Weaknesses:**

STRENGTHS
======
1) The idea seems simple while quite effective. For the reported case, the experiments are convincing.
2) Figures captions are insightful, and I do not have many problems to understand the paper, even if I am not an expert in the field.

WEAKNESSES
=====
1) I find the addressed problem interesting, but I would suggest providing more insights on the applicative relevance of the paper. What is a potential use case? E.g., navigate through models with similar overall performances but with different quality on specific cases?
2) Due to the paper space limits, experiments are quite limited, which is entirely understandable. I think experiments with different models and in different contexts would significantly strengthen the finding. For example, would performing tests on language models or other tasks, like generation/regression, be possible?
3) I would love to have more insights into the models' interpretability and performance of the algorithm. How different are the obtained models along the path? What is the computational cost of the procedure?

**Suggested Changes:**

I would suggest addressing the above weakness as follows:
1) Adding a paragraph about applications, detailing potential use cases, and showing an example for the proposed method.
2) Adding at least another experiment on a different data modality or task, providing qualitative examples and analysis on the behaviour of models along the geodesic path.
3) Elaborating a bit more on the future works and the conclusions; for example, I would find it interesting to incorporate further constraints (e.g., find networks with similar performance but also handling classes unbalanced) or network pruning (e.g., check how weights change along the geodesic path and analyze differences in weights with higher and lower variance). I would also add a discussion on the eventual limitations of the work.
4) Including analysis of the computational cost of the approach. How long does it take for convergence?

---

### Official Review · Reviewer_sb7A · 2023-04-02

**Confidence:** 4

**Summary Of Contributions:**

The paper studies the mode connectivity if the neural network. This paper proposed understanding and achieving mode connectivity from an information geometry perspective.

**Rating:**

High Impact (HI): a submission which meets the reviewing criteria and is predicted to make an impact on the field

**Strengths And Weaknesses:**

1. This paper is interesting. Casting the mode-connecting paths to geodesics is interesting.
2. This paper also proposed an algorithm to approximate geodesics. And the experimental results in Figure 2 clearly show the effectiveness of the proposed method.

I like this paper and do find weaknesses.


**Suggested Changes:**

N/A

---

### Author Response · Authors · 2023-05-01
**Response to Reviewers**

We thank the reviewers and area chair for their insightful feedback, and are encouraged by the positive response this work received.

We are particularly grateful for the detailed suggestions of Reviewer tP1F. Given the space constraints of the TinyPaper format, a longer format submission is in preparation. We feel suggestions #1, #2 and #3 are best addressed in this follow-up work. We will address suggested change #4 in an appendix.

We once again thank the reviewers for their consideration of our work.

---

### Author Response · Authors · 2023-05-30
**Opt-in for archival**

The authors wish to opt-in for archival in the ICLR Tiny Papers track.

---

### Meta-Review · Area_Chair_q7oa · 2023-04-07

**Recommendation:** Invite to present (notable)
**Confidence:** 4

**Metareview:**

This paper is clear, concise, and interesting. The reviewers agree that it meets the Clear, Correct, and Reproducible (CCR) standards. It is additionally quite insightful and the proposed method effective.

**Summary:**

This paper studies mode connectivity in the context of geodesics in distribution space.

**Reason For Not Giving A Higher Recommendation:**

NA

**Reason For Not Giving A Lower Recommendation:**

The reviewers and I find no substantial defects in the paper in the scope of this venue.

---

### Decision · Program_Chairs · 2023-04-09

Invite to present (notable)